# Multimodal personalised executive function intervention (E-Fit) for school-aged children with complex congenital heart disease: protocol for a randomised controlled feasibility study

Alenka Sarah Schmid [1], Melanie Ehrler [1,2,3], Flavia Wehrle [1,2,4], Ruth O'Gorman Tuura,[1,2,5,6] Oliver Kretschmar,[2,5,7] Markus Landolt [2,8,9] Beatrice Latal[1,2,3,5]

**Correspondence to**
Dr Beatrice Latal;
bea.latal@kispi.uzh.ch

## ABSTRACT

**Introduction** Children with congenital heart disease (CHD) are at risk for executive functions (EF) impairments. To date, interventions have limited effects on EF in children and adolescents with complex CHD. Therefore, we developed a new multimodal and personalised EF intervention (E-Fit). This study aims to test the feasibility of this intervention called 'E-Fit' for children with complex CHD and EF impairments.

**Methods and analysis** This is a single-centre, single-blinded, randomised controlled feasibility study exploring the E-Fit intervention. We aim to enrol 40 children with CHD aged 10–12 years who underwent infant cardiopulmonary bypass surgery and show clinically relevant EF impairments (T-score ≥60 on any Behaviour Rating Inventory for Executive Function questionnaire summary scale). The multimodal intervention was developed with focus groups and the Delphi method involving children and adolescents with CHD, their parents and teachers, and health professionals. The intervention is composed of three elements: computer-based EF training using CogniFit Inc 2022, performed three times a week at home; weekly EF remote strategy coaching and analogue games. The content of the computer and strategy training is personalised to the child's EF difficulties. The control group follows their daily routines as before and completes a diary about their everyday activities four times a week. Participants will be randomised in a 1:1 ratio. Feasibility is measured by the participants' and providers' ratings of the participants' adherence and exposure to the intervention, recruitment rates and the evaluation of the intended effects of the programme.

**Ethics and dissemination** Local ethics committee approval was obtained for the study (BASEC-Nr: 2021-02413). Parents provide written informed consent. Key outputs from the trial will be disseminated through presentations at conferences, peer-reviewed publications and directly to participating families. Furthermore, these results will inform the decision whether to proceed to a randomised controlled trial to investigate effectiveness.

**Trial registration number** NCT05198583.

## STRENGTHS AND LIMITATIONS OF THIS STUDY

⇒ The intervention was designed based on extensive input from patients with congenital heart disease, families, teachers and health professionals provided through focus group interviews and the Delphi method.

⇒ Using a multiple-domain approach, the intervention will be personalised towards individual patients' and families' needs.

⇒ Assessors are blinded to the group allocation of participants.

⇒ Families of the control group may be aware of the content of the intervention and engage in its activities, which might reduce the effect sizes of the intervention.

⇒ Considering the limited sample size of this feasibility study, the effectiveness of the training will only be estimated exploratorily.

## INTRODUCTION

Eight out of 1000 children are born with congenital heart disease (CHD)[1]; of these, one-third require open heart surgery during infancy.[2] In these children, neurodevelopmental problems are known to be a main comorbidity.[3] Studies have shown that the mean IQ of children with CHD is within the normal range but lower than that of their healthy peers.[4] Despite this normal intellectual functioning, children with complex CHD, particularly, are at risk for higher-order cognitive deficits: executive function (EF) impairments.[4 5] The main EF are inhibition; cognitive flexibility, including fluency; planning and working memory.[6–10] Children with CHD show mild to moderate impairments in all these EF domains[4 5] without a clear predominance of any specific EF domain. In typically developing children, EF are more

closely related to school readiness than IQ, and EF predict academic achievement.[11] Accordingly, EF difficulties in childhood have been related to higher unemployment rates in adulthood.[12] Further, EF difficulties in adulthood can increase the risk of addictive behaviours and psychiatric diseases such as anxiety and depression.[13 14]

Many sequelae of EF difficulties have also been demonstrated in children with CHD. They face lower school achievement and lower psychosocial quality of life (QoL) during school age.[15 16] Further, lower EF have been associated with a higher rate of internalising symptoms in adolescents and adults with CHD.[17] Given the lifelong academic, psychological and social burden associated with EF difficulties, evidence-based interventions improving EF are urgently needed, particularly for children and adolescents with CHD.[18 19]

## EF intervention

To date, only two studies have investigated the efficacy of EF interventions in children with CHD.[20 21] Both studies included school-aged children and adolescents with CHD and tested Cogmed Working Memory Training (Cogmed[22]). Cogmed is a computerised programme designed to improve attention and working memory, a core dimension of EF.[22] Participants completed regular Cogmed sessions at home for at least 5 weeks.[20 21] These studies demonstrated inconsistent findings: Calderon *et al* found no improvement of working memory either at postintervention or at 3-month follow-up, but participants who did the Cogmed sessions showed significant improvements in inhibitory control and self-regulation at the 3-month follow-up. Parents reported improved cognitive regulatory skills at postintervention and 3-month follow-up.[21]

In contrast, Jordan *et al* found improved working memory postintervention in participants who were assigned to Cogmed compared with the control group. However, this improvement did not persist longer than 6 months follow-up.[20] These findings are in line with the inconsistent results of a recent meta-analysis of EF interventions in other children at risk for EF impairments.[23] This meta-analysis reported only small to moderate effects of computerised interventions of single functions and no evidence for long-term effects. These inconsistencies between EF interventions have led to investigations about factors impacting the effectiveness of an EF intervention.[23] One of those factors is motivation, which influences the effort a participant is willing to make, which further impacts the intervention's success.[24] Game elements are able to increase motivation, and therefore, using computerised interventions with game elements is a favourable option.[25] Also, analogue games, such as board and card games, have been tested as EF interventions and have shown effectiveness in enhancing the functions trained.[26 27] Moreover, by requiring children to play with others, such games may enhance self-monitoring.[26] Studies combining computer and analogue games demonstrated improvements in trained reasoning

and speed tasks in healthy children with low socioeconomic status (SES) aged 7–10 years.[27] However, despite the enjoyment factor, a major challenge in interventional studies is the limited transfer of training effects to other cognitive abilities.[28–31] Importantly, a meta-analysis revealed that programmes emphasising teaching social-emotional strategies and self-regulation had the most significant positive impact on EF, particularly for children with neurodevelopmental disorders or behavioural problems. These effects appeared to be stronger than those observed in trainings primarily focusing on tasks specifically training EF and may address the transfer of training to general cognitive abilities.[18 28 32] Therefore, exploring intervention approaches beyond computerised and tasks-specific training is crucial.[18] Mindfulness education has improved EF behaviour, optimism and classroom social competence behaviours in healthy children and adolescents.[33 34] Similarly, problem solving solutions to EF-related challenges in everyday life improved parent-rated and self-rated global EF behaviours in children with epilepsy.[32] Hence, mindfulness interventions and strategy coaching may address the transfer of training to everyday life by targeting sensory awareness and attention regulation, thus promoting emotional and social development.[18 32–34]

Combining established and engaging computerised and analogue training that improves individual EF, alongside strategy coaching aiming at integrating these benefits into daily life, presents a promising approach for addressing the aforementioned limitations of current interventions.

Because children with CHD experience difficulties in various EF domains with significant academic, psychological and social consequences, an intervention integrating several of these approaches may be particularly suited to improving these children's EF and, eventually, their long-term development and social integration.

## Aims and hypotheses

This study aims to test the feasibility of a multimodal personalised EF intervention for children with complex CHD and EF impairments. To evaluate E-Fit's feasibility, the following aims and hypotheses will be explored:

Aim 1: Assess whether children with CHD and EF impairments can adhere to E-Fit.

Hypothesis 1: Study participants will complete at least 80% of the strategy coaching and computerised training sessions of E-Fit.

Aim 2: Assess the acceptability of E-Fit for children with CHD and EF impairments and their parents.

Hypothesis 2: Study participants and their parents will indicate high acceptability of E-Fit as measured by average ratings of 3 or higher on a 4-point rating scale assessing the acceptability of each programme component.

Aim 3: Assess the feasibility of E-Fit for children with CHD and EF impairments and their parents.

Hypothesis 3: Study participants and their parents will indicate high feasibility of E-Fit as measured by average

ratings of 3 or higher on a 4-point rating scale assessing the user-friendliness and ease of integrating each programme component into their daily lives and environments.

Aim 4 (exploratory): Measure the impact of E-Fit on improving EF skills in children with CHD and EF impairments.

Hypothesis 4: We anticipate a moderate effect in the direction of improved EF from preintervention to postintervention in study participants.

The rationale for projection numbers provided in the hypotheses is as follows:

Adherence of 80% is based on previous intervention studies.[21]

The decision to use rating scale ratings of 3 or higher is founded on the 4-point rating scale (1=do not agree at all, 4=fully agree). Therefore, 3 and above are considered affirmative.

Effect sizes are based on the results of a meta-analysis. Specifically, a value of 0.38 pertains to explicit EF practice (including computerised and analogue training across all populations), while 0.46 corresponds to strategy coaching across all populations.[23]

## METHODS AND ANALYSIS

In this section, we will first describe the development of the intervention, which involved a needs assessment preceding the present investigation. Subsequently, we will outline the methods used in the present feasibility study. For an overview, see figure 1.

### Needs assessment

Before developing E-Fit and this study, we conducted a needs assessment. This assessment aimed to specifically identify the unique EF needs of children with CHD by means of focus group interviews and the Delphi method. Below, you will find comprehensive details about the focus group interviews and the Delphi method, including the outcomes that led to the final intervention.

### Focus group interviews

We conducted two separate online focus group interviews: one with five adolescents with CHD aged 14–16 years and another with seven of their parents ($n_{female}$=5). Participants were recruited from the Teen Heart Study, which investigated EF in children and adolescents with CHD at the University Children's Hospital Zurich. The Teen Heart Study included children and adolescents with CHD without genetic or dysmorphic syndromes who were born between 2004 and 2012 and underwent their first cardiopulmonary bypass (CPB) surgery before the age of 6 years.[35] Participants were asked what they consider the ideal content and scope of an intervention supporting EF in children with CHD (eg, (1) how much time can or should your child invest per (a) week, (b) per day, (c) how long in total? (2) Which difficulties does your child have in everyday life?; see online supplemental file 1). These interviews were recorded with the consent of the participants and then transcribed and analysed with structured qualitative content analysis.[36] This analysis produced nine categories, and all participants' suggestions and statements were assigned to one of these categories (online supplemental table S1).

### Delphi method

The results from the focus group interviews informed the initiation of the Delphi method, aiming to achieve consensus on the intervention's content through three sequential questionnaires.[37] During this procedure, the items of a survey are rated and evaluated multiple times, each time anonymously. The participants subsequently receive the next survey, along with feedback on the overall answers.[38 39]

The surveys were completed online by children with CHD (n=7) aged 12–13 years who had not participated in the preceding focus group interviews, their teachers (n=5), parents (n=7) and health professionals (n=5). The health professionals were experts in the field from international hospitals and research institutions who held a comprehensive understanding of the difficulties faced by children with CHD. They had previously engaged with the research group and were contacted by email. The families were again recruited from the Teen Heart Study, but following focus group interview suggestions, only participants between 10 and 13 years old were asked to participate. The families provided the email addresses of the teachers to be contacted by the study team. Hence, the surveys were completed by four stakeholder groups: children with CHD, parents of children with CHD, teachers of children with CHD, and health professionals.

For survey 1, respondents received a link by email and were invited to rate each item on a Likert scale from 1 for 'fully disagree' to 9 for 'fully agree'. They were encouraged to make further suggestions yet to be represented in the survey. Survey 2 was sent to all respondents of survey 1. All items from survey 1 were included, and additional items suggested by participants were added following revision and categorisation. The results of survey 1 were included in survey 2: A table displayed the median and IQR per stakeholder group for each item, and each participant was reminded of the scores they had given in survey 1. Participants were asked to consider the input of all stakeholder groups in survey 1 and then review their answers. Items reaching consensus (IQR<2) were dropped from the subsequent round. Survey 3 was sent to all respondents of survey 2. Items for which no consensus was reached in survey 2 were included. No more items were added. Again, the results of survey 2 were included in survey 3. Participants again reviewed their answers. At the end of survey 3, every participant was asked about their interest in participating in the final consensus meeting. Throughout the process, all stakeholder groups responded to the same items.

Overall, participants completed three rounds with the goal of consensus-seeking across stakeholder groups, defined as IQR<2 for each item.[40] Items for which no

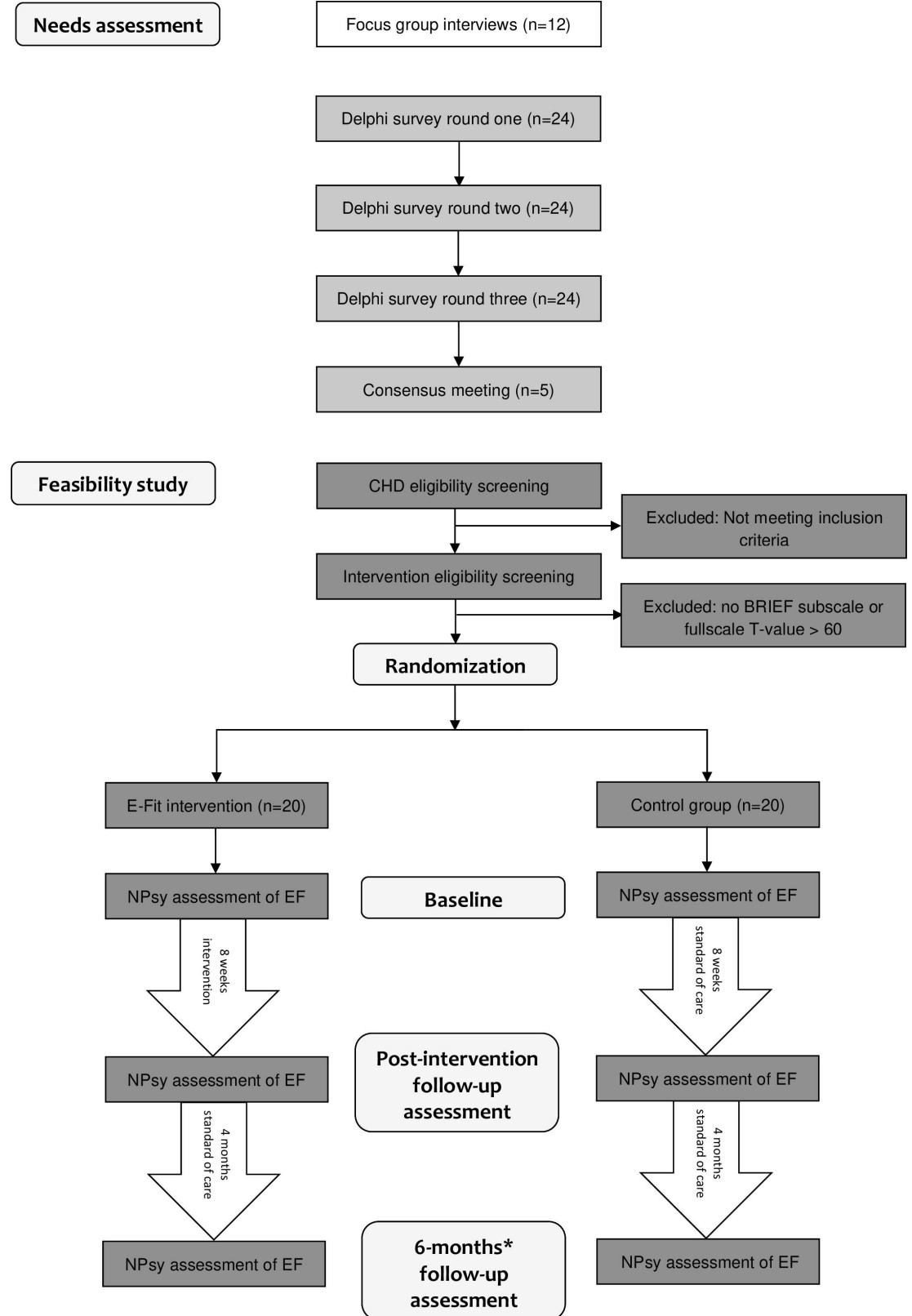

**Figure 1** Flow chart of intervention development and trial design. *from baseline. BRIEF, Behaviour Rating Inventory of Executive Functions; CHD, congenital heart disease; EF, executive function; E-Fit, EF intervention.

consensus was reached after three rounds were discussed in a final consensus meeting with three parents, two health professionals and the study team. In the consensus meeting, it became clear that some aspects of the

intervention are child-specific and cannot be answered conclusively. To families, it was particularly important to have a diverse intervention that was not too school-like. The researchers commented on the intervention in terms

of methodological aspects, which were taken into account in the study design (eg, additional questionnaires or incentives to be added).

## Findings from needs assessment

The results of the Delphi method, including the consensus meeting, showed that the wishes most strongly expressed by parents were that the intervention should include strategies supporting homework and everyday life activities and that the children should learn to develop their own strategies. Families participating in the Delphi method wanted activities that did not resemble schoolwork, improved reasoning skills, and addressed academic future planning (online supplemental figure S1). Parents emphasised that the ideal age at which an EF intervention should start depends on the individual child, ranging from 4 to 18. Literature shows that EF development progresses considerably during childhood and adolescence.[18] This opens a window of opportunity for EF training during these periods.[41 42] Also, the families' perceived need for therapy is crucial for adherence.[43] During school age and adolescence, families become more aware of the sequelae of CHD and thus become more willing to participate in an intervention programme. Participants also wanted minimal parental participation in the intervention, and the focus group interviews indicated that participants became aware of EF problems shortly before high school when remedial action might still be effective.

Consequently, we decided on an age range of 10–12 years instead of younger or older. No consensus was reached on whether the intervention should include personal meetings with the study team, whether the intervention should occur in a nearby community centre, or on the design of the games (online supplemental figure S1). The consensus meeting indicated that responses on these topics differed between children and families, and general agreement was difficult to reach. Hence, literature and practice in previous programmes were used to decide on these elements.[18 20 21 43–48] For details, see online supplemental file 2 and supplemental material online supplemental figure S2.

## Study design to test the feasibility of the intervention

This is a single-centre, single-blinded, randomised controlled feasibility study to test the feasibility of the E-Fit intervention.

All participants undergo a baseline neurodevelopmental assessment (table 1) and are randomly assigned to either the E-Fit intervention or to the control group. All participants undergo a 2-month postintervention and a 6-month follow-up assessment. Investigators administering neuropsychological tests are blinded to the participant's allocation. The families and the psychologist who conducts the intervention are aware of the allocation (online supplemental table S2).

**Table 1** Executive function domains

| EF domain | Assessment test |
| --- | --- |
| Inhibition | Colour Word Interference Task (D-KEFS[70]) Go/No-Go (TAP[71]) |
| Cognitive flexibility | Trail-making Test (D-KEFS[70]) Colour Word Interference Task (D-KEFS[70]) |
| Planning | Tower Test (D-KEFS[70]) |
| Working memory | Digit Span (WISC-V[66]) Picture Span (WISC-V[66]) No Letter Sequencing (WISC-V[66]) |
| Risk-taking | Balloon Analogue Risk Task[72] |

D-KEFS, Delis-Kaplan Executive Function System; EF, executive function; WISC-V, Wechsler Intelligence Scale for Children fifth edition.

## Participants and recruitment

This study includes children with CHD who underwent CPB in Switzerland before 1 year of age. Diagnoses may include transposition of the great arteries, hypoplastic left heart syndrome and other complex CHDs requiring CPB. At recruitment, children are 10–12 years old. Children diagnosed with a genetic or dysmorphic syndrome such as trisomy-21 or Ellis-van Creveld syndrome or with known large cerebral lesions or injuries, including stroke and severe hypoxic-ischaemic encephalopathy with significant neurological sequelae preventing them from performing the assessments are excluded.

Recruitment for the study commenced in April 2022 and is planned to continue until fall 2023. Children from the Department of Cardiology outpatient clinic at the University Children's Hospital Zurich are being contacted consecutively. Follow-up assessments are scheduled to be completed by the beginning of 2024. Participants are identified through the clinical database comprising all treated children with CHD. All eligible children with CHD and their parents receive a letter with the study information and are asked to contact the study team if they are interested in participating. The study team provides further information about the study and answers questions on the phone. The contact information of the children's teachers is forwarded by the parents, and after agreeing to participate and giving written informed consent (online supplemental file 3), parents and the child's principal teacher fill in the Behaviour Rating Inventory of Executive Functions (BRIEF[49]) screening questionnaire. The BRIEF assesses EF behaviours in children and adolescents on three summary scales: a Behaviour Regulation Index, a Metacognition Index and a Global Executive Composite. Participants are included in the study if a T-score≥60 on any of the three summary scales is reported by either parents or teachers.

## Randomisation and stratification

After passing the screening questionnaire, participants will be randomly assigned to either the intervention or control group in a 1:1 design by block randomisation (block size=4, stratification variable: maternal education level on a 6-point Likert scale). This list of random numbers will be generated prior to the start of the study in the statistical software R[50] and implemented in the randomisation module on the research electronic database Redcap,[51] which allows allocation to be concealed until the participant is enrolled. At the end of the first assessment, after all tests and questionnaires are completed, the child will open an envelope prepared by a study member otherwise not involved in the study. The note in the envelope will reveal the allocation status to the family and baseline assessor.

## Intervention group: E-Fit

E-Fit targets multiple EF domains, focusing on four that have been identified as being impaired by two meta-analyses on EF in children with CHD: inhibition, planning, flexibility and working memory.[4 5] See online supplemental file 4 for details. These are addressed in a multimodal approach consisting of three personalised and adaptive elements: (1) computerised training played alone, (2) remote 1:1 strategy coaching with the child and a psychologist and (3) analogue board and card games fostering EF abilities that can be played alone or with friends and family (online supplemental table S3). The intervention lasts for 8 weeks.

### Computerised training

Participants use the platform CogniFit Inc 2022 (CogniFit) for the computerised training. CogniFit was selected based on literature[52 53] and its successful use in other similarly aged populations[54] and because it targets several EF domains. This is in contrast to Cogmed, which only targets working memory. From the CogniFit platform, 11 games targeting the aforementioned EF domains were selected, considering 3 games per function (online supplemental table S4). These games involve tasks that require information retention and manipulation, strategy development and task switching.[55] These games are adaptive to the child's level of performance. Participants play the assigned games at home on a computer, smartphone or tablet. A device is provided if needed. The child is asked to play three times a week for 20 min over the course of the 8-week intervention period.

### Strategy coaching

The strategy coaching is conducted remotely. A psychologist is assigned to each family as a coach to monitor and support them throughout the intervention. The child meets with the coach once a week for 60 min for a total of eight sessions.

The first session discusses baseline assessment results and the child's strengths and weaknesses.[56] Further, the first module of CogniFit is performed to make sure the child understands how to use CogniFit. In this session, a parent participates with the child.

For the six following sessions, parents fill in a short questionnaire[47] to identify EF domains in which their child faces difficulties. Of 11 EF domains rated, the 6 indicating the most difficulties are selected for the coaching sessions. Each of these six sessions has standardised guidelines and start with a story about an EF and an explanation of how this function engages our brain.[44–46 48] Afterwards, the child completes a task teaching a strategy for the EF in question[47] (for more details, see online supplemental file 4). During the coaching sessions, solutions to difficulties with adhering to E-Fit are suggested.

The eighth and final coaching session consists of the wrap-up and outlook, where a parent again participates with the child.[57]

Additionally, the coach calls the parents in the first and fourth week to discuss the upcoming sessions that will be conducted with the child only. Parents can suggest specific situations to be addressed in the subsequent strategy coaching. In response, the coach provides tips on how parents can support their child with the respective EF.[47] For an overview of the strategy coaching procedure, see table 2.

Following the manual, the strategy coaching teaches strategies to promote EF in everyday life.[47] A clinical neuropsychologist reviewed the strategy coaching, and the manual is available in German language.

### Analogue training

Thirteen board and card games are provided to the families (online supplemental table S5). They are suited for this age group and target EF abilities. A special needs and

| Table 2 | Strategy coaching procedure | | | | | | | |
|---------|---------|---------|---------|---------|---------|---------|---------|---------|
| | Week 1 | Week 2 | Week 3 | Week 4 | Week 5 | Week 6 | Week 7 | Week 8 |
| Child | | 60 min individual topic | 60 min individual topic | 60 min individual topic | 60 min individual topic | 60 min individual topic | 60 min individual topic | |
| Parent | 30 min call | | | 30 min call | | | | |
| Parent and child | 60 min initialisation session | | | | | | | 45 min wrap up session |

trained game educator helped select the games targeting the above-mentioned EF domains for children between 10 and 12 years of age. Like the computerised training, the analogue training also demands a range of diverse functions, including rule retention, flexible adaptation to other players, impulse control and strategic thinking. Often, a single game may require multiple functions simultaneously. Families are free to play as much as they want but are asked to report the date, duration and name of the analogue games played.

### Control group

The control group receives standard of care, including cardiac surveillance, neurodevelopmental counselling and therapies such as speech- and physiotherapy. Because the control group families know the scope of the tested intervention, this may lead to contamination: during the trial, parents of control children may engage their children in activities used in the intervention fostering EF. To detect this and where occurring, to measure the time spent on it, parents and children complete an electronic diary four times a week for 8 weeks to monitor the child's everyday activities (online supplemental file 5 and online supplemental table S3). After the final visit at the end of study participation, the control group families receive the list of EF-enhancing analogue games used, EF fact sheets and a 2-month CogniFit login.

### Primary outcome measures

The feasibility of the E-Fit intervention will be measured in six areas, as described by Bowen *et al*.[58] Except for Practicality and Limited efficacy, primary outcome measures will be collected only in the intervention group throughout the intervention period. For a summary of measures and assessment time points, see table 3. The following areas will be assessed.

#### Acceptability

Acceptability examines suitability, satisfaction and attractiveness to programme deliverers and recipients. It is measured by questions about satisfaction, intent to continue use and perceived appropriateness throughout the intervention period.

#### Demand

Demand is assessed by the extent to which the intervention is likely to be used or how much demand exists. This includes measures for the fit within the family, perceived positive or negative effects on the family, actual use, expressed interest or intention to use, and perceived demand throughout the intervention period.

#### Implementation

To ensure the successful delivery of the intervention to participants, implementation is a critical aspect. The E-Fit Fidelity Measurement System was newly developed following existing guidelines.[59] The purpose of this system is to measure the degree of fidelity with which the newly developed E-Fit was implemented. Essential components, including specific questions to be asked and required session completion, were organised, phrased and complemented by response choices (yes and no). During the study, the Fidelity Measurement System will be piloted and evaluated.

#### Practicality

Practicality explores how an intervention can be delivered to intended participants with existing means, resources and circumstances. It includes factors affecting implementation ease or difficulty, efficiency, speed and quality of implementation, positive or negative effects on target participants, and the participant's ability to complete the intervention activities throughout the intervention period.

#### Integration

Integration assesses the extent to which the intervention can be integrated within the existing system. This addresses the perceived fit with infrastructure and perceived sustainability during the last coaching session.

#### Limited efficacy

Limited efficacy explores whether the new intervention shows promise of succeeding with the intended population. This is measured by evaluating the programme's intended effects or process on key intermediate variables, effect-size estimation and maintenance of changes from initial change (see table 1 for an overview of EF measures).

### Secondary outcome measures

In addition to primary outcome measures, we will examine other variables assessing the success of the intervention. The following secondary outcome measures are to be collected from both groups at baseline, 2-month and 6-month follow-up: The Family Relationship Index is a self-report measure used to evaluate subjective quality of family environment for parents.[60] The Kidscreen-10 assesses the child's subjective health and well-being, and the Kidscreen-27 assesses the child's health and well-being from a parent's perspective; these are combined to assess the child's QoL.[61] The parents will complete the short form of the Conners-3 and the Social Responsiveness Scale[62 63] to screen for behavioural problems. The Resilience Scale (RS-13) is a self-reported questionnaire assessing resilience by acceptance of self-competencies, life competencies and personal competencies. This questionnaire will be completed by the parents and the child.[64]

### Covariate measures

Medical, sociodemographic and patient-specific factors and IQ will be measured in both groups at baseline and be treated as covariates for the effectiveness of the intervention and to describe the population who participated in the intervention.

SES will be estimated on a six-point scale for both paternal and maternal education, leading to the lowest possible SES sum score of 2 and the highest of 12.[65] Cardiac and neonatal data will be collected from medical

**Table 3** Feasibility focus areas and assessment tools

| Area of focus | Assessment tool | Group | Time point | Completed by |
|---|---|---|---|---|
| Acceptability | Acceptance and Feasibility Scale[32] | E-Fit | Final coaching | Parents and children |
| | Children's Feeling Scale[73] | E-Fit | Intervention period, before and after each online training | Children |
| | Observed engagement[74] | E-Fit | Each coaching session | Coach |
| Demand | Acceptance and Feasibility Scale[32] | E-Fit | Final coaching | Parents and children |
| | Fun and Demand[75] | E-Fit | Intervention period, before and after each online training | Children |
| | Session data: duration, no of completed sessions and for the computerised training, the coaching and the card games | E-Fit | Intervention period | |
| | Recruitment rate | E-Fit | End of study | |
| Implementation | E-Fit Fidelity Measurement System tool, developed based on Feely *et al*[59] | E-Fit | End of study | Two separate raters |
| Practicality | Acceptance and Feasibility Scale[32] | E-Fit | Final coaching | Parents and children |
| | Feedback in coaching sessions | E-Fit | Each coaching session | Parents and children |
| | Observed involvement[74] | E-Fit | End of study | |
| | Session rating scale[76] | E-Fit | Each coaching session | Children |
| | Observed engagement[74] | E-Fit | Each coaching session | Coaches |
| | Recruitment rates | Both | End of study | |
| | Retention rates | Both | End of study | |
| | Completion of weekly sessions | E-Fit | End of study | |
| | No of adverse events | Both | Continuously | |
| Integration | Acceptance and Feasibility Scale | E-Fit | Final coaching | Parents and children |
| Limited efficacy | Delis-Kaplan Executive Function System[70] | Both | BL, $T_1$, $T_2$ | |
| | Wechsler Intelligence Scale for Children- fifth edition[66] | Both | BL, $T_1$, $T_2$ | |
| | Test of Attentional Performance[71] | Both | BL, $T_1$, $T_2$ | |
| | Balloon Analogue Risk Task[77] | Both | BL, $T_1$, $T_2$ | |
| | Behaviour Rating Inventory for Executive Function | Both | BL, $T_1$, $T_2$ | Parents* and teachers |

*If possible, completed by both parents separately.
E-Fit, executive function intervention.

records. A short form of the Wechsler Intelligence Scale for Children, fifth edition, will be used to assess total IQ once at the baseline assessment.[66]

## Sample size determination

This is a proof-of-concept study to test the feasibility of a new intervention. Accordingly, the primary outcome is feasibility; thus, no minimum sample size has been calculated.[67 68] A retrospective power analysis will be conducted. Instead, the sample size was defined from previous literature.[68 69] Using a pilot study with a sample size <30 may lead to the potential to under-recruit for the main study.[68] To provide a high level of confidence, a total sample size between 35 and 55 is suggested to account also for small effect sizes. As the total population to recruit from for this study is relatively small, we decided on a sample size of 40 (E-Fit intervention group n=20, control group n=20).

Approximately 210 children are eligible for screening and are to be contacted. In a study by Spillmann et al[15] using a cut-off of >65 in the BRIEF, around 20% of children with complex CHD had EF difficulties at 10 years. Using data from children and adolescents between 10 and 15 years and a cut-off of >60, we estimate around 30% of eligible children to demonstrate EF difficulties, leading to a potential sample size of 60 children. We assume a response rate of around 75% from those families with children with EF problems, leading to a study sample of approximately 45 participants. A drop-out rate of approximately 10% allows the planned sample of 40 participants to be achieved.

## Data analysis plan

The feasibility of the intervention will be delineated from the descriptive statistics.

Descriptive statistics will be calculated, including means, SD, medians and IQRs for continuous variables and frequency counts and percentages for categorical variables. The newly developed E-Fit Fidelity Measurement System will be evaluated. Two raters will complete it individually, and inter-rater reliability will be reported. To report limited efficacy, within-subject differences between EF performances from baseline to post-treatment and 6-month follow-up assessments are to be investigated across treatment groups with an intention-to-treat analysis. Linear regression controlling for covariates, involving differences in means and 95% CI, will be used to assess differences between groups and time points for continuous outcomes. Proportions and logistic regressions for binary outcomes will be used to examine group effects (eg, comparison of clinically relevant vs not clinically relevant scores in the BRIEF, $\chi^2$ tests, ORs and 95% CI). Given that missing values are missing at random, they will be imputed using multiple imputations.

The data will be visually inspected to identify discordant and influential observations. All analyses will be conducted with an alpha level of 0.05. The statistical software R will be used for all analyses.[50] The final project

report will describe and justify any deviations from the original statistical plan.

## Patient and public involvement

As described above, children and adolescents with CHD, their parents, and their teachers were involved in the intervention development and design. Study results will be disseminated to participants and their families through a newsletter and a public informative meeting.

## ETHICS AND DISSEMINATION

The protocol was approved by the ethical committee of the Canton of Zurich in Switzerland (BASEC-Nr: 2021-02413) and was registered on ClinicalTrials.gov (NCT05198583). Written informed consent will be obtained from the parents. Data handling, record keeping and archiving will be performed according to the guidelines given by the ethical committee's guidelines (for details, see online supplemental file 6). The results of this study will be disseminated through presentations at national and international conferences, publications in peer-reviewed journals, and directly to families who participated in the study.

Key outputs from the trial will contribute to our dissemination and impact agenda: Pilot data will inform the decision whether to proceed to a definitive randomised controlled trial, and if we do, we will have evidence to support funding applications and a finalised intervention delivery manual to enable multicentre replication of E-Fit.

## DISCUSSION

This article outlines the development and design of a randomised controlled feasibility study investigating an 8-week EF intervention for children with CHD. Given the limited effects of previous EF interventions, E-Fit was newly developed based on comprehensive input from patients, families, teachers and health professionals. It includes strategy coaching and both computerised and analogue game-based training and is personalised to the individual participant.

This study evaluates feasibility through the areas of focus acceptability, demand, implementation, practicality, integration and limited efficacy. The results of this study will lay the foundation for a main randomised controlled trial to investigate effectiveness in a larger sample. While E-Fit was initially developed to address EF deficits in children with CHD, successful feasibility validation could also broaden the applicability of E-Fit to other paediatric populations facing EF difficulties.

Limitations of the study include that families of the control group may be aware of the content of the intervention and engage in its activities, which might reduce the effect sizes of the intervention. Additionally, considering the limited sample size of this feasibility study,

the effectiveness of the training will only be estimated exploratorily.

## Author affiliations

[1]Child Development Center, University Children's Hospital Zurich, Zurich, Switzerland
[2]Children's Research Center, University Children's Hospital Zurich, Zurich, Switzerland
[3]URPP Adaptive Brain Circuits in Development and Learning, University of Zurich, Zurich, Switzerland
[4]Department of Neonatology and Intensive Care, University Children's Hospital Zurich, Zurich, Switzerland
[5]University of Zurich, Zurich, Switzerland
[6]MR Research Centre, University Children's Hospital Zurich, Zurich, Switzerland
[7]Department of Cardiology, University Children's Hospital Zurich, Zurich, Switzerland
[8]Department of Psychosomatics and Psychiatry, University Children's Hospital Zurich, Zurich, Switzerland
[9]Division of Child and Adolescent Health Psychology, Department of Psychology, University of Zurich, Zurich, Switzerland

**Acknowledgements** The authors thank Nina Zeltner, who supported the qualitative analysis and Thomas Pletschko, who reviewed the structure and content of the strategy coaching.

**Contributors** BL, ME, FW, RO'GT, OK, ML and ASS conceived and designed the study. BL is the grant holder. ASS wrote and revised the manuscript. All authors critically revised the manuscript for important intellectual content.

**Funding** This work was supported by the Swiss National Science Foundation (SNF) grant number 32003B_172914.

**Disclaimer** This funding source had no role in the design of this study and will not have any role during its execution, analyses, interpretation of data, or decision to submit results. CogniFit 2022 had no role in the design of this study and will not have any role during its execution, analyses, interpretation of data, or decision to submit results.

**Competing interests** None declared.

**Patient and public involvement** Patients and/or the public were involved in the design, or conduct, or reporting, or dissemination plans of this research. Refer to the Methods section for further details.

**Patient consent for publication** Consent obtained from parent(s)/guardian(s).

## ORCID iDs

Alenka Sarah Schmid http://orcid.org/0000-0001-5013-0530
Melanie Ehrler http://orcid.org/0000-0003-4253-2016
Flavia Wehrle http://orcid.org/0000-0001-5992-0424
Markus Landolt http://orcid.org/0000-0003-0760-5558

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
