## [Reviewer comments · BMJ Open]

ARTICLE DETAILS

TITLE (PROVISIONAL)	Multimodal personalized executive function intervention (E-Fit) for school-age children with complex congenital heart disease: protocol for a randomized controlled feasibility study
AUTHORS	Schmid, Alenka; Ehrler, Melanie; Wehrle, Flavia; Tuura, Ruth; Kretschmar, Oliver; Landolt, Markus; Latal, Beatrice

VERSION 1 – REVIEW

REVIEWER	Brossard-Racine, Marie RI-MUHC
REVIEW RETURNED	11-May-2023

GENERAL COMMENTS	Thank you for the opportunity to review this manuscript. Overall, this is an important study that has used a patient-oriented research approach to develop the intervention to be tested in a feasibility study. Although the information provided in the document is excellent and sufficient, I would think the manuscript could reach a greater audience and be more impactful if the authors would consider re-organizing the presentation of their extensive work. To my understanding, the current proposal includes two studies: 1.) a patient-oriented intervention development study (focus group, Delphi, review of the literature) and 2.) a feasibility study (RCT). I believe that the manuscript would be clearer and more digestible if these 2 sections were presented separately with their respective sets of objectives, hypotheses, and methodological sections: participants, analyses etc... The following comments I have listed reflect my search for clarification. Abstract: I am confused by the goal of the current proposal. Is it a planning study vs. a feasibility study or an intervention study? Perhaps consider rephrasing a clear aim at the end of the introduction using similar wording used in section 1.2. Strength and limitation: First statement: the novelty of the proposed intervention is a bit redundant as phrased: "first study to assess the feasibility of a novel..." I think the novelty aspect to be amplified here may be the multimodal and personalized aspect of the intervention (- not so much that is it the first feasibility study...). Consider replacing "patients" with participants, individuals, children, adolescents etc... where present throughout the document. The rationale for exploring other interventions than computer games could be stronger. While the evidence from previous trials
---

	is well summarized, the flow of ideas in the last two paragraphs of section 1.1 is a bit choppy. The statements regarding the effectiveness of board games and motivation could be more strongly developed and supported with references. The aim in section 1.2 is clear, however, some of the hypotheses are a bit difficult to conceptualize. For instance, what is the difference between hypotheses 1 and 3 (i.e., adhere to all components vs. can integrate all components)? Are they projection numbers that could perhaps be integrated? E.g., % of satisfaction or % of compliance? Methods and analyses A flow chart of the different steps (interviews, different surveys, final meeting etc..) would be useful. 2.1: is the need assessment done by the two focus groups? Please consider rewording for clarity. 2.1.1 Are the questions for the interview available? Or perhaps a few concrete examples could be provided in the text or join the interview guide in appendices? 2.1.2 There is a need for the different surveys to be articulated with what is being referred to as “the Delphi survey” in the first paragraph. When is the “final consensus meeting” taking place? Is it part of 2.1.3? If so I would advise using similar terminology. Were the revised versions of the surveys the same for all 4 groups of stakeholders or were these separate surveys? The rest of the information provided in this section is clear and sufficient. 2.1.3 I am struggling with the headline of the section “Final suggestions” are these the results/consensus from the Delphi procedure? If so, perhaps this could be reflected in the headline. 2.3 I wonder if the presentation of the participants could be presented at the beginning of the methods section and by doing so could lighten the subsequent section that refers to the participants. Grammar: Future tense is used in this paragraph while previous sections are in the past tense Information on the recruitment of the children is clear, however, it is not clear how parents, teachers and health professionals were enrolled for the planning study. Statement on patient involvement (2.4) may be better placed toward the end of the documents along with ethical consideration. Intervention is overall clearly described and so are the primary and secondary outcome measures. However, I have found myself looking for information regarding baseline assessment when first introduced in the text as well as regarding what questionnaire is being used to determine the child’s EF difficulties. Perhaps the table of outcomes could be introduced earlier for clarity. I do understand that a major component of the provided intervention is the “personalized” intervention that is being provided and that is met to be tailored to the child’s difficulties. However, considering that EF encompasses such a broad range of abilities that are sometimes highly related to emotional regulation (hot EF) vs. higher-order cognitive processes (cold EF), I wonder if
--	--

	the intervention is considering both cold and hot aspects of EF or one more than the other. Perhaps a few additional sentences on the content of the provided intervention would clarify this.
--	--

REVIEWER	Robledo Castro, Carolina Universidad del Tolima, Departamento de pedagogía y mediaciones tecnológicas
REVIEW RETURNED	09-Jul-2023

GENERAL COMMENTS	In general, the protocol shows methodological rigor and clarity in the procedures to be carried out. Here are some suggestions to take into account: The article does not include the dates of the study. It is important to add these dates in the manuscript. In the methodology session, there is a sub-session called Intervention development, however I found it a bit confusing for the reader, at first they state: "we conducted a needs assessment to address the unique EF needs of children with CHD", but the content of the following bullets seem like loose ideas. I suggest to the authors that in the introductory paragraph of "Intervention development" they describe more precisely what this sub-session is about, what is the methodological process that they are going to present and what they are going to present in the following bullets. The authors point out that the BRIEF scale will be used as a criterion for inclusion in the study. This is self-explanatory, but in the analysis plan the authors state that BRIEF scale data will be used to examine group effects. In this sense, I consider it important in the secondary measures section to present and describe the BRIEF scale and the variable that is expected to be measured with this instrument. I suggest to include the CONSORT checklist.
--

REVIEWER	Bunge, Silvia University of California Berkeley, Psychology
REVIEW RETURNED	10-Jul-2023

GENERAL COMMENTS	This is a very thoughtfully designed intervention program aimed at improving executive functions (EFs) in children, specifically focused on children with congenital heart disease. This intervention improves on what is already available, in multiple ways: 1) it was designed based on extensive input from patients, families, teachers, and health professionals; 2) it will tackle EFs in multiple ways: strategy coaching and both computerized and analog games, marrying two approaches that are typically not combined; and 3) it will be tailored towards individual patients' and families' needs. The supplementary materials provide many details regarding the development and planned implementation of the intervention. The outcome measures and questionnaires have been thoughtfully selected/constructed. Main comments: I would broaden the framing of the protocol; rather than calling it an intervention for children with CHD, I would call it a tailored, multi-pronged EF intervention -- and note that it's been developed for
---

	treatment of EF deficits in children with CHD, but that it could be applied to other pediatric populations with EF deficits. That would increase the target audience for this paper. However, this is just a suggestion; I am sure there are good arguments in favor of sticking with the current framing. I am concerned about the possibility that 8 weeks will not be sufficient for durable, widespread changes in child outcomes. The duration may need to be reconsidered after piloting. I am encouraged that most families were willing to participate in a longer intervention. A few more details in the main text would be helpful for readers who don't delve into the supplementary materials. In particular:  1. The main text should specify that the intervention is designed to take place three times a week for an overall duration of 8 weeks, and should provide an overview of the amount of time to be dedicated to each portion (recognizing that this may not yet be known for the analog games). 2. I recommend provide a very brief description of the Delphi survey procedure -- what the method is, broadly speaking -- before diving into the details of how it was implemented in this protocol. 3. Say a little bit more on p. 11 about the board and card games - i.e., the fact that they are each theorized to target one or more EF skills, as indicated in the table. Will any guidance be provided re: how long/how often children/families should try to play them (or is it up to them in this initial feasibility study? Can you say anything about how the games were selected? Note that the link to this PDF isn't working: https://www.spielendfoerdern.ch/assets/spielliste_1.5.pdf What about indicating, as a new column in the table, which games are played individually and/or in a group? Minor points: B.1: Some of the text here is redundant with what is in the main text. Typo in Figure S1, third-to-last row: "indepently" Several of the tables/plots in the supplement appear to be duplicated towards the end of the PDF.
--	---

REVIEWER	Wolfe, Kelly
REVIEW RETURNED	13-Jul-2023

GENERAL COMMENTS	Thank you for the opportunity to review this interesting and exciting study protocol. While a multitude of studies demonstrate the prevalence and impact of executive functioning (EF) deficits in individuals with congenital heart disease, there are very few research-based interventions to treat these difficulties. More intervention research is needed for this issue; the present manuscript reports on a thoughtfully designed EF intervention protocol and a study designed to examine the efficacy of this intervention. Introduction:
---

	Pg 6, line 21: Please provide citation(s) for the statement that analog games have been shown to improve EFs such as problem-solving. Aims: Each of the 4 hypotheses would benefit from increased specificity about how success will be defined. For example, in #1, is the hypothesis that 100% of participants in the intervention group will complete 100% of the sessions? For #2, what is the hypothesized or goal aggregate satisfaction level for the program (e.g., a mean satisfaction score of 3+ on a scale of 1-4 where 4 is completely satisfied)? Methods & Analysis: Focus group: It would be helpful to report information about the individuals who participated in the two focus groups (e.g., # of individuals, basic demographic information). Table S1 is referenced as noting the 9 categories derived from the focus group, but Table S1 appears to be a study timeline. Please review all table & figure references in the body of the protocol and ensure the in-text references match the numbering in the supplement. Intervention Group: The description of “computerized training alone” is a bit misleading as no participants are only receiving computerized training without the coaching and analog games components; would suggest removing “alone” and/or adding “coaching and analog games” for clarity. Primary Outcome Measures: It would be helpful to include a brief description of the E-Fit Fidelity Measurement System that was developed for this study (page 12, line 17) Supplemental Materials: Please review the supplemental tables and figures for duplication (for example, pages 29-30 and 49-50 appear to show the same figures), and ensure they appear in order (S1, S2, S3, etc).
--	--

VERSION 1 – AUTHOR RESPONSE

Reviewer: 1

Dr. Marie Brossard-Racine, RI-MUHC Comments to the Author:

Thank you for the opportunity to review this manuscript. Overall, this is an important study that has used a patient-oriented research approach to develop the intervention to be tested in a feasibility study. Although the information provided in the document is excellent and sufficient, I would think the manuscript could reach a greater audience and be more impactful if the authors would consider re-organizing the presentation of their extensive work. Response: We thank Dr. Marie Brossard-Racine for providing valuable feedback on our manuscript. We appreciate her thoughtful suggestions, expertise and attention to detail, which have significantly contributed to improving the overall clarity and organization of the paper.

To my understanding, the current proposal includes two studies: 1.) a patient-oriented intervention development study (focus group, Delphi, review of the literature) and 2.) a feasibility study (RCT). I believe that the manuscript would be clearer and more digestible if these 2 sections were presented separately with their respective sets of objectives, hypotheses, and methodological sections: participants, analyses etc... The following comments I have listed reflect my search for clarification.

Response: After careful consideration, we have concluded that restructuring the manuscript in the suggested manner may not enhance clarity as intended. The patient-oriented intervention development discussed in the manuscript was not conducted as a separate study per se; rather, it served as the foundation for the subsequent intervention study. As such, we believe that retaining the current structure effectively highlights the role of the patient-oriented development intervention as a basis for the subsequent study.

Abstract: I am confused by the goal of the current proposal. Is it a planning study vs. a feasibility study or an intervention study? Perhaps consider rephrasing a clear aim at the end of the introduction using similar wording used in section 1.2.

Response: We thank the reviewer for this suggestion. We have added appropriate wording at the end of the introduction section in the abstract as suggested.

Please see changes in the manuscript:

Abstract (page 2): "This study aims to test the feasibility of this intervention called "E-Fit" for children with complex CHD and EF impairments."

Strength and limitation:

First statement: the novelty of the proposed intervention is a bit redundant as phrased: "first study to assess the feasibility of a novel..." I think the novelty aspect to be amplified here may be the multimodal and personalized aspect of the intervention (- not so much that is it the first feasibility study...).

Response: We acknowledge the point you raised regarding this strength, which we initially included in the manuscript. However, upon receiving the editor's comment, the point did not directly address a methodological aspect, as required. Consequently, we removed it to ensure compliance with the editorial guidelines. Please refer to the manuscript to see the new list of strength and limitations.

Consider replacing "patients" with participants, individuals, children, adolescents etc... where present throughout the document.

Response: We have made the revisions accordingly. We chose to use the terms "children," "children and adolescents," and "participants" alternatively. Please see changes directly in the manuscript.

The rationale for exploring other interventions than computer games could be stronger. While the evidence from previous trials is well summarized, the flow of ideas in the last two paragraphs of section 1.1 is a bit choppy. The statements regarding the effectiveness of board games and motivation could be more strongly developed and supported with references.

Response: The details about the effectiveness of board games was previously mentioned only in the supplementary section. We have now incorporated it into the main text. Furthermore, we have made an effort to revise the section 1.1 where we discuss the significance of motivation and to delve more extensively into the exploration of alternative interventions.

Please see changes in the manuscript:

Section 1.1 (page 5): “In contrast, Jordan et al. found improved working memory postintervention in participants who were assigned to Cogmed compared to the control group. However, this improvement did not persist longer than 6-month follow up [21]. These findings are in line with the inconsistent results of a recent meta-analysis of EF interventions in other children at risk for EF impairments [24]. This meta-analysis reported only small to moderate effects of computerized interventions of single functions and no evidence for long-term effects. These inconsistencies between EF interventions have led to investigations about factors impacting the effectiveness of an EF intervention [25]. One of those factors is motivation, which influences the effort a participant is willing to make, which further impacts the intervention’s success [26]. Game elements are able to increase motivation and therefore, utilizing computerized interventions with game elements are a favorable option [27]. Also, analog games, such as board and card games have been tested as EF interventions and have shown effectiveness in enhancing the functions trained [28,29]. Moreover, by requiring children to play with others, such games may enhance self-monitoring [28]. Studies combining computer and analog games demonstrated improvements in trained reasoning and speed tasks in healthy children with low socioeconomic status aged 7 to 10 years [30]. However, despite the enjoyment factor, a major challenge in interventional studies is the limited transfer of intervention effects to other cognitive abilities [31–34]. Importantly, a metaanalysis revealed that programs teaching social-emotional strategies and self-regulation had the most significant positive impact on EF, particularly for children with neurodevelopmental disorders or behavioral problems. These effects appeared to be stronger than those observed in interventions primarily focusing on tasks specifically training EF. [18,24,31]. Therefore, exploring intervention approaches beyond computerized and tasks specific training is crucial [20]. Problem-solving solutions to EF-related challenges in everyday life improved parent- and self-rated global EF behaviors in children with epilepsy [37]. Hence, strategy coaching may address the transfer of intervention to everyday life by targeting sensory awareness and attention regulation, thus promoting emotional and social development [31].

Combining established and engaging computerized and analog interventions that improve individual EF, alongside strategy coaching aiming at integrating these benefits into daily life, presents a promising approach for addressing the aforementioned limitations of current interventions.”

The aim in section 1.2 is clear, however, some of the hypotheses are a bit difficult to conceptualize. For instance, what is the difference between hypotheses 1 and 3 (i.e., adhere to all components vs. can integrate all components)? Are they projection numbers that could perhaps be integrated? E.g., % of satisfaction or % of compliance?

Response: Based on your suggestions, we have reformulated the hypotheses, incorporating projection numbers to provide more concrete statements. Additionally, we have taken care to define technical terms, such as "integrate," in a clear and concise manner to avoid any ambiguity.

Please see changes in the manuscript:

Section 1.2 (pages 5-6)

- “Children with CHD and EF impairments adhere to at least 80% of the strategy coaching and computerized training sessions.
- Children with CHD and EF impairments and their parents find all components of the intervention suitable, enjoyable and report a high level of satisfaction, all rated as 3 or higher on a 4-point rating scale.
- Children with CHD and EF impairments can integrate all components of the E-Fit intervention into their lives and environment, as it is designed to be user-friendly and easily accessible, all rated as 3 or higher on a 4-point rating scale.

- Although exploratory due to its small sample size, we anticipate a moderate effect size (.38 to .46) of the E-Fit intervention for improving EFs in children with CHD and EF impairments pre- to post-intervention.

The rationale for projection numbers provided in the hypotheses is as follows:

Adherence of 80% is based on previous intervention studies [23].

The decision to use rating scale ratings of 3 or higher is founded on the 4-point rating scale (1 = do not agree at all - 4 fully agree), Therefore, 3 and above are considered affirmative. Effect sizes are based on results of a meta-analysis. Specifically, a value of .38 pertains to explicit EF practice (including computerized and analog training, across all populations), while .46 corresponds to strategy coaching across all populations [24].”

Methods and analyses

A flow chart of the different steps (interviews, different surveys, final meeting etc..) would be useful.

Response: We agree and have created a flow chart (figure 1, page 19) to enhance the clarity of the manuscript. There were two focus groups, one with parents and one with children with CHD. For the Delphi method another set of families, teachers and health professionals were recruited.

2.1: is the need assessment done by the two focus groups? Please consider rewording for clarity.

Response: We have taken note of the clarity issue and reworded the sentence to explicitly state that both the focus groups and the Delphi method were part of the needs assessment process, and also refer to the needs assessment throughout the manuscript. For both parts, the focus groups and Delphi method, different participants were recruited.

Please see changes in the manuscript:

Section 2.1 (page 6): "Before developing E-Fit and the present study, we conducted a needs assessment. This assessment aimed to specifically identify the unique EF intervention needs of children

with CHD by means of focus group interviews and the Delphi method. Below, you will find comprehensive details about the focus group interviews and Delphi method, including the outcomes that led to the final intervention.” Section 2.1.3 (page 8): “2.1.3 Findings from needs assessment”

2.1.1 Are the questions for the interview available? Or perhaps a few concrete examples could be provided in the text or join the interview guide in appendices?

Response: We have included concrete examples in the text to illustrate our procedure. Furthermore, we added the interview guide (translated from German into English) in the appendices (1_Appendix).

Please see changes in the manuscript:

Section 2.1.1 (page 6): “Participants were asked what they consider the ideal content and scope of an intervention supporting EFs in children with CHD (e.g.: 1. how much time can or should your child invest a) per week, b) per day, c) how long in total? 2. Which difficulties does your child have in everyday life?, see 1_Appendix).”

2.1.2 There is a need for the different surveys to be articulated with what is being referred to as “the Delphi survey” in the first paragraph.

Response: Thank you for your feedback regarding the confusion in our terminology. All three of the surveys involved in our research are classified as Delphi surveys. To improve clarity, we have now renamed the entire procedure as “the Delphi method”, and refer to the individual Delphi surveys just as “surveys”. Also see figure 1, page 19 for more clarity.

Please see changes in the manuscript:

Section 2.1.2 (page 7): “2.1.2 Delphi method

The results from the focus group interviews informed the initiation of the Delphi method, aiming to achieve consensus on the intervention’s content through three sequential questionnaires [27,28]. During this procedure, the items of a survey are rated and evaluated multiple times, each time anonymously. The participants subsequently receive the next survey, along with feedback on the overall answers [29].”

When is the “final consensus meeting” taking place? Is it part of 2.1.3? If so I would advise using similar terminology.

Response: The final consensus meeting was scheduled to take place after the third round of the Delphi method. We have rephrased paragraph 2.1.3 and also added a visualization of the Delphi method in figure 1, page 19, to ensure greater clarity.

Please see changes in the manuscript:

Section 2.1.3 (page 8): “The results of the Delphi method, including the consensus meeting, showed that the wishes most strongly expressed by parents were that the intervention should include strategies supporting homework and everyday life activities and that the children should learn to develop their own strategies.”

Were the revised versions of the surveys the same for all 4 groups of stakeholders or were these separate surveys? The rest of the information provided in this section is clear and sufficient.

Response: Thank you for addressing this unclarity. The revised versions were the same for all 4 stakeholder groups. We have added this information to the manuscript.

Please see changes in the manuscript:

Section 2.1.2 (page 7): "At the end of survey three, every participant was asked about their interest in participating in the final consensus meeting. Throughout the process, all stakeholder groups responded to the same items."

2.1.3 I am struggling with the headline of the section "Final suggestions" are these the results/consensus from the Delphi procedure? If so, perhaps this could be reflected in the headline.

Response: Thank you for pointing this out. We changed the title of section 2.1.3 (page 8) to "Findings from needs assessment". This section includes the outcomes of the focus groups and the Delphi method, which serve as the foundation for the development of the intervention.

2.3 I wonder if the presentation of the participants could be presented at the beginning of the methods section and by doing so could lighten the subsequent section that refers to the participants.

Response: Thank you for your comment. We would like to clarify that the participants described in paragraph 2.3 are not the same individuals who participated in the Delphi method. Instead, they are the children who will be recruited for the intervention itself. We understand the importance of providing accurate information about the participants, and we hope that the added flow chart (figure 1) makes it better understandable.

Grammar: Future tense is used in this paragraph while previous sections are in the past tense

Response: As you correctly pointed out, in some paragraphs, we used past tense, particularly when addressing the needs assessment, which preceded the intervention. We agree with your assessment that the protocol describes an ongoing study. Hence, we have decided to use the present simple tense in this specific section to accurately reflect the current status of the research.

Information on the recruitment of the children is clear, however, it is not clear how parents, teachers and health professionals were enrolled for the planning study.

Response: Thank you for pointing this out. To ensure a comprehensive recruitment process, we simultaneously contact the parents along with the participants through a combined approach of letter invitations and phone calls. Additionally, the parents play an integral role in providing us with the contact information of the teachers, this information was added to the manuscript. Health professionals were only recruited for the Delphi method. They were known experts in the field, who were known to the research group and were contacted by email.

Please see changes in the manuscript:

Section 2.1.2 (page 7): "The health professionals were experts in the field from international hospitals and research institutions, who held a comprehensive understanding of the difficulties faced by children with CHD. They had previously engaged with the research group and were contacted by email. [...] The families provided the e-mail addresses of the teachers to be contacted by the study team."

Section 2.3 (page 9): "The contact information of the children's teachers are forwarded by the parents and after agreeing to participate and giving written informed consent (3_Appendix) parents and the child's principal teacher fill in the Behavior Rating Inventory of Executive Functions (BRIEF [40]) screening questionnaire."

Statement on patient involvement (2.4) may be better placed toward the end of the documents along with ethical consideration.

Response: The journal's requirements state that the paragraph should indeed be included in the methods section. However, we agree with your observation that its initial placement might not have been suitable and have relocated the paragraph to the end of the methods section.

Intervention is overall clearly described and so are the primary and secondary outcome measures. However, I have found myself looking for information regarding baseline assessment when first introduced in the text as well as regarding what questionnaire is being used to determine the child's EF difficulties. Perhaps the table of outcomes could be introduced earlier for clarity.

Response: Thank you for your valuable comment. As the primary focus is on feasibility, we decided not to separately include a section about the baseline assessment. To address the challenge of locating relevant information, we have introduced the dedicated table 1 (page 17) for EF outcomes earlier in section 2.2. Further, we have added the citation to the screening questionnaire upon its initial mention in the text.

Please see changes in the manuscript:

Section 2.2 (page 8): "All participants undergo a baseline neurodevelopmental assessment (table 1) and are then randomly assigned to either the E-Fit intervention or to the control group."

I do understand that a major component of the provided intervention is the "personalized" intervention that is being provided and that is met to be tailored to the child's difficulties. However, considering that EF encompasses such a broad range of abilities that are sometimes highly related to emotional regulation (hot EF) vs. higher-order cognitive processes (cold EF), I wonder if the intervention is considering both cold and hot aspects of EF or one more than the other. Perhaps a few additional sentences on the content of the provided intervention would clarify this.

Response: Indeed, the intervention encompasses both hot and cold EF, and we have listed the potential domains of EF from which a subset is selected for each child. While it is true that cold EF are more explicitly targeted in this intervention, we did not introduce these specific terms in the paper, as the intervention was not designed explicitly based on these domains. Also, the degree of targeted hot and cold EF varies from child to child, as it is a tailored intervention. Nonetheless, your comment emphasized the importance of better defining the focus and domain of the EF targeted in our study. To address this, we specified the content of the intervention in the manuscript, which covers both types of EF, however, we chose not to explicitly label them as "hot" or "cold" EF.

Please see changes in the manuscript:

Section 2.5 (page 10): "This is in contrast to Cogmed, which only targets working memory. From the CogniFit platform, eleven games targeting the aforementioned EF domains were selected considering three games per function (table S4). These games involve tasks that require information retention and manipulation, strategy development and task switching."

Section 2.5 (page 11): "A special needs and trained game educator helped select the games targeting the above-mentioned EF domains for children between 10 to 12 years of age. Like the computerized training, the analog training also demands a range of diverse functions, including rule retention, flexible adaptation to other players, impulse control, and strategic thinking. Often, a single game may require multiple functions simultaneously."

4_Appendix Section B (page 1): "Possible topics are working memory, attention control, emotion regulation, flexibility, initiating action, metacognition, organization, planning, response inhibition, time management and goal-directed persistence."

Reviewer: 2

Dr. Carolina Robledo Castro, Universidad del Tolima Comments to the Author:

In general, the protocol shows methodological rigor and clarity in the procedures to be carried out. Here are some suggestions to take into account:

Response: We thank Dr. Carolina Robledo Castro for her valuable feedback on our manuscript. We genuinely value the time and effort she dedicated to evaluating our manuscript, and are grateful for her contributions

The article does not include the dates of the study. It is important to add these dates in the manuscript.

Response: Thank you for this comment. We have included the requested information in the 'Participants and recruitment' section.

Please see changes in the manuscript:

Section 2.3 (page 9): "Recruitment for the study commenced in April 2022 and is planned to continue until fall 2023. Children from the Department of Cardiology outpatient clinic at the University Children's Hospital Zurich are being contacted consecutively. Follow-up assessments are scheduled to be completed by the beginning of 2024."

In the methodology session, there is a sub-session called Intervention development, however I found it a bit confusing for the reader, at first they state: "we conducted a needs assessment to address the unique EF needs of children with CHD", but the content of the following bullets seem like loose ideas. I suggest to the authors that in the introductory paragraph of "Intervention development" they describe more precisely what this sub-session is about, what is the methodological process that they are going to present and what they are going to present in the following bullets.

Response: Thank you for your comment. In response to your comment, we have revised the first paragraph to provide clearer guidance on what readers can expect from the subsequent bullet points.

Please see changes in the manuscript:

Section 2 (page 6): "In this section, we will first describe the development of the intervention, which involved a needs assessment preceding the present investigation. Subsequently, we will outline the methods utilized in the present feasibility study. For an overview, see figure 1."

The authors point out that the BRIEF scale will be used as a criterion for inclusion in the study. This is self-explanatory, but in the analysis plan the authors state that BRIEF scale data will be used to examine group effects. In this sense, I consider it important in the secondary measures section to present and describe the BRIEF scale and the variable that is expected to be measured with this instrument.

Response: Thank you for this important comment. In fact, the BRIEF scale is not only used as an inclusion criterion tool, it is also assessed at the post-intervention and 6-months followup as part of the limited efficacy, which is a primary outcome measure. It is hence described in table 3 (page 18). The primary outcomes are described in a table due to their comprehensiveness.

I suggest to include the CONSORT checklist. [NOTE FROM THE EDITORS: This last point should be rebutted, the authors have correctly used the SPIRIT checklist for protocols, CONSORT is intended to guide the reporting of trial results]

Response: We appreciate your attention to reporting guidelines. In fact, the CONSORT checklist is intended to guide the reporting of trial results, however, because we are presenting a protocol for the study, the journal requires us to use the SPIRIT (Standard Protocol Items: Recommendations for Interventional Trials) checklist for reporting protocols.

Reviewer: 3

Silvia Bunge

Comments to the Author:

This is a very thoughtfully designed intervention program aimed at improving executive functions (EFs) in children, specifically focused on children with congenital heart disease. This intervention improves on what is already available, in multiple ways: 1) it was designed based on extensive input from patients, families, teachers, and health professionals; 2) it will tackle EFs in multiple ways: strategy coaching and both computerized and analog games, marrying two approaches that are typically not combined; and 3) it will be tailored towards individual patients' and families' needs. The supplementary materials provide many details regarding the development and planned implementation of the intervention. The outcome measures and questionnaires have been thoughtfully selected/constructed. Response: We thank Prof. Silvia Bunge for her valuable feedback on our manuscript. Her thorough review of our manuscript, insightful feedback and constructive comments have greatly enriched the quality of our work.

Main comments:

I would broaden the framing of the protocol; rather than calling it an intervention for children with CHD, I would call it a tailored, multi-pronged EF intervention -- and note that it's been developed for treatment of EF deficits in children with CHD, but that it could be applied to other pediatric populations with EF deficits. That would increase the target audience for this paper. However, this is just a suggestion; I am sure there are good arguments in favor of sticking with the current framing.

Response: Thank you for your valuable suggestion regarding the framing of our protocol. The development of this intervention has been concentrated on children with CHD, aiming to address EF deficits in this specific population. We acknowledge that there may be opportunities to extend the application of this intervention to other pediatric populations with EF deficits in the future. We agree with your point that discussing the potential for broader use is valuable and have added the section 4 Discussion, where we address the potential to broaden the intervention to other pediatric populations.

Please see changes in the manuscript:

Section 4 (page 15): 4. Discussion

This article outlines the development and design of a randomized-controlled feasibility study, investigating an 8-week EF intervention called "E-Fit" for children with CHD. Given the limited effects of previous EF interventions, E-Fit was newly developed based on comprehensive input from patients, families, teachers, and health professionals. It includes a strategy coaching, and both computerized and analog game-based training, and is personalized to the individual participant.

This study evaluates feasibility through the areas of focus acceptability, demand, implementation, practicality, integration and limited efficacy. The results of this study will lay the foundation for a main randomized controlled trial to investigate effectiveness in a larger sample. While E-Fit was initially developed to address EF deficits in children with CHD, successful feasibility validation could also broaden the applicability of E-Fit to other pediatric populations facing EF difficulties.

I am concerned about the possibility that 8 weeks will not be sufficient for durable, widespread changes in child outcomes. The duration may need to be reconsidered after piloting. I am encouraged that most families were willing to participate in a longer intervention.

Response: Thank you for your important insight. The decision on the duration of the study was based on previous literature and research in the field. However, we fully acknowledge the exploratory nature of this feasibility study. As such, we are mindful of the need to remain open to potential adjustments

and refinements in various aspects, including the study duration. The Feasibility and Acceptability questionnaires also include questions regarding the satisfaction with the duration. Upon completing the feasibility study, we will thoroughly analyze the results and consider any pertinent questions that arise. This will allow us to make informed decisions and further refine our approach for subsequent phases of the research.

A few more details in the main text would be helpful for readers who don't delve into the supplementary materials. In particular:

1. The main text should specify that the intervention is designed to take place three times a week for an overall duration of 8 weeks, and should provide an overview of the amount of time to be dedicated to each portion (recognizing that this may not yet be known for the analog games).

Response: We appreciate your thoughtful suggestions, which have helped us enhance the clarity and comprehensiveness of the paper. We have now specified in the main text that the intervention is designed to take place three times a week, with a total duration of 8 weeks. The duration of the coaching can be found in section 2.5 (page 10).

Please see changes in the manuscript:

Section 2.5 (page 10): "The child is asked to play three times a week for 20 min over the course of the eight-weeks intervention period."

2. I recommend provide a very brief description of the Delphi survey procedure -- what the method is, broadly speaking -- before diving into the details of how it was implemented in this protocol.

Response: We have now included a short introduction to the Delphi method, to give readers a broader understanding of the procedure.

Please see changes in the manuscript:

Section 2.1.2 (page 7): "The results from the focus group interviews informed the initiation of the Delphi method, aiming to achieve consensus on the intervention's content through three sequential questionnaires [27,28]. During this procedure, the items of a survey are rated and evaluated multiple times, each time anonymously. The participants subsequently receive the next survey, along with feedback on the overall answers [29]."

3. Say a little bit more on p. 11 about the board and card games - i.e., the fact that they are each theorized to target one or more EF skills, as indicated in the table.

Response: This point has also been raised by another reviewer, and we have now expanded on the description of the board and card games. Specifically, we now include information about how the games are theorized to target one or more EF skills, as indicated in the tables S4 and S5 (supplemental material (pages 4 to 5)).

Please see changes in the manuscript:

Section 2.5 (page 11): "Like the computerized training, the analog training also demands a range of diverse functions, including rule retention, flexible adaptation to other players, quick reactions, and strategic thinking. Often, a single game may require multiple functions simultaneously."

Will any guidance be provided re: how long/how often children/families should try to play them (or is it up to them in this initial feasibility study)?

Response: At this stage, participants and their families have complete freedom in deciding how long and how often they would like to engage with the analog games. We will observe their natural patterns of usage during the feasibility study to gain insights into the practicality and acceptability of this component of the intervention. We added this information to the manuscript.

Please see changes in the manuscript:

Section 2.5 (page 11): "Families are free to play as much as they want, but are asked to report the date, duration, and name of the analog games played."

Can you say anything about how the games were selected?

Response: Regarding the selection of games for the intervention, we have taken into consideration multiple factors:

The online games, were chosen based on the specific EF they target, as indicated by the information provided by the Cognifit platform. The Cognifit platform provides games which are adaptive to the skills level of the participants and increase in difficulty, as the participant progresses, and we targeted each executive function with three games.

For the analog games, a special needs and trained game educator consulted the study team in the choice of games. The targeted EF can be found in an online games list, which is also run by her and her team (www.spielendoernern.ch). The suitability of the analog games for the participants' age group was a key consideration, and an effort was made to include some games that can be played individually. This ensures that participants have the flexibility to engage in the intervention without necessarily depending on other players, thus enhancing accessibility and usability.

Details about the games can be found in the tables S4 and S5 (supplemental material (pages 4 to 5)).

Please see changes in the manuscript:

Section 2.5 (page 10): "From the CogniFit platform, eleven games targeting the aforementioned EF domains were selected considering three games per function (table S4). These games involve tasks that require information retention and manipulation, strategy development and task switching. [34]"

Section 2.5 (page 11): "A special needs and trained game educator, helped select the games targeting the above-mentioned EF domains for children between 10 to 12 years of age. Like the computerized training, the analog training also demands a range of diverse functions, including rule retention, flexible adaptation to other players, impulse control, and strategic thinking. Often, a single game may require multiple functions simultaneously."

Note that the link to this PDF isn't working:

Response: Thank you for bringing this to our attention. Upon your notification, we replaced the link accordingly:

https://www.spielendoernern.ch/_files/ugd/8fa6b3_f84a167cfa3c405f8d3244a83242e7ac.p df

What about indicating, as a new column in the table, which games are played individually and/or in a group?

Response: Thank you for this suggestion. In response, we have now incorporated a "players" column in table S5 (supplemental material (page 5)).

Minor points:

B.1: Some of the text here is redundant with what is in the main text.

Response: Thank you for your comment. We recognized that the specific part you mentioned was already adequately covered in the main text and have removed this section from the appendix (2_Appendix B.1 (page 1)).

Typo in Figure S1, third-to-last row: "indepently"

Response: Thank you for pointing out the error in the plot. We have corrected the typo accordingly.

Several of the tables/plots in the supplement appear to be duplicated towards the end of the PDF.

Response: Thank you for taking the time to review the figures and tables in our manuscript. We conducted a thorough review and cross-referencing of all figures and tables to ensure there are no repetitions or duplications.

Reviewer: 4

Kelly Wolfe

Comments to the Author:

Thank you for the opportunity to review this interesting and exciting study protocol. While a multitude of studies demonstrate the prevalence and impact of executive functioning (EF) deficits in individuals with congenital heart disease, there are very few research-based interventions to treat these difficulties. More intervention research is needed for this issue; the present manuscript reports on a thoughtfully designed EF intervention protocol and a study designed to examine the efficacy of this intervention.

Response: We thank Kelly Wolfe, PhD for her valuable feedback on our manuscript. Her thoughtful suggestions have helped us to refine the study further, and we are confident that her guidance will lead to an improved final publication.

Introduction:

Pg 6, line 21: Please provide citation(s) for the statement that analog games have been shown to improve EFs such as problem-solving.

Response: Thank you for your comment. We have provided citations for the statement regarding the effectiveness of analog games in improving executive functions.

Please see changes in the manuscript:

Section 1.1 (page 5): "Also, analog games, such as board and card games have been tested as cognitive interventions and have shown effectiveness in enhancing the functions trained [28,29]. Moreover, by requiring children to play with others, such games may enhance selfmonitoring [28]."

28 Gonçalves PD, Ometto M, Sendoya G, et al. Neuropsychological Rehabilitation of Executive Functions: Challenges and Perspectives. *J Behav Brain Sci* 2014;2014. doi:10.4236/jbbs.2014.41004

29 Estrada-Plana V, Esquerda M, Mangues R, et al. A Pilot Study of the Efficacy of a Cognitive Training Based on Board Games in Children with Attention-Deficit/Hyperactivity Disorder: A Randomized Controlled Trial. *Games Health J* 2019;8:265–74. doi:10.1089/g4h.2018.0051

Aims:

Each of the 4 hypotheses would benefit from increased specificity about how success will be defined. For example, in #1, is the hypothesis that 100% of participants in the intervention group will complete

100% of the sessions? For #2, what is the hypothesized or goal aggregate satisfaction level for the program (e.g., a mean satisfaction score of 3+ on a scale of 1-4 where 4 is completely satisfied)?

Response: Thank you for bringing this issue to our attention. Another reviewer also highlighted the same point. We have increased the specificity of how the verification of our hypotheses will be defined in our study. By including specific numbers and criteria that will be used to accept or reject the hypotheses.

Please see response to reviewer 1, page 5, and changes in the manuscript in section 1.2 (pages 5-6).

Methods & Analysis:

Focus group: It would be helpful to report information about the individuals who participated in the two focus groups (e.g., # of individuals, basic demographic information).

Response: As an anonymous needs assessment, we did not gather additional demographic information beyond age, sex and the number of participants. In response to your comment, we have included the number and sex of individuals in the focus group, while detailed demographic data remains unavailable.

Please see changes in the manuscript:

Section 2.1.1 (page 6) “We conducted two separate online focus group interviews: one with five adolescents with CHD aged 14 to 16 years and another with seven of their parents (n_{female}= 5).”

Table S1 is referenced as noting the 9 categories derived from the focus group, but Table S1 appears to be a study timeline. Please review all table & figure references in the body of the protocol and ensure the in-text references match the numbering in the supplement.

Response: Thank you for bringing the inconsistency in table referencing to our attention. We have revised the table referencing to ensure accuracy throughout the manuscript.

Intervention Group: The description of “computerized training alone” is a bit misleading as no participants are only receiving computerized training without the coaching and analog games components; would suggest removing “alone” and/or adding “coaching and analog games” for clarity.

Response: We intended to state that the children play the online games “alone”. However, we understand that the insertion of "alone" in this context may be misleading. We have revised the sentence to read "played alone" instead.

Primary Outcome Measures: It would be helpful to include a brief description of the E-Fit

Fidelity Measurement System that was developed for this study (page 12, line 17) Response: Thank you for your comment. In response to your suggestion, we have now included a brief description of the E-Fit Fidelity Measurement System in the manuscript. Also, the E-Fit Fidelity Measurement System can be found in the Appendix 5, pages 6 to 10.

Please see changes in the manuscript:

Section 2.7 (page 12): “To ensure the successful delivery of the intervention to participants, implementation is a critical aspect. The E-Fit Fidelity Measurement System was newly developed following existing guidelines [50]. The purpose of this system is to measure the degree of fidelity with which the newly developed E-Fit was implemented. Essential components, including specific questions to be asked and required session completion, were organized, phrased and complemented by response choices (yes and no). During the study, the Fidelity Measurement System will be piloted and evaluated.”

Supplemental Materials: Please review the supplemental tables and figures for duplication (for example, pages 29-30 and 49-50 appear to show the same figures), and ensure they appear in order (S1, S2, S3, etc).

Response: In response to your feedback, we reviewed the manuscript and removed the duplicated tables and figures.

To address the supplementary details requested by the reviewers, we have revised the abstract to ensure compliance with the prescribed word constraints. Also, for consistency, we have replaces all EFs by EF.

VERSION 2 – REVIEW

REVIEWER	Brossard-Racine, Marie RI-MUHC
REVIEW RETURNED	01-Sep-2023

GENERAL COMMENTS	The revised version read smoothly and the authors have addressed all my previous comments. This is very exciting work and I look forward to seeing the results of this study.
---

REVIEWER	Bunge, Silvia University of California Berkeley, Psychology
REVIEW RETURNED	26-Aug-2023

GENERAL COMMENTS	The authors have addressed all of my concerns.
--

REVIEWER	Wolfe, Kelly 06-Sep-2023
REVIEW RETURNED	

GENERAL COMMENTS	Thank you for the opportunity to review this revised study protocol. The authors have done an excellent and thorough job addressing the points raised by other reviewers and myself. I have one area of requested editing for this revised draft which is to further hone and refine the Specific Aims. Having very clearly defined Aims and Hypotheses will be crucial for evaluating the study's future results. The authors did re-work some of the Aims in response to my first revision, but the Aims should be refined further to focus only on empirical or measurable outcomes, and the fourth Aim is currently written as a hypothesis. I think it would be most useful to include both Aims and Hypotheses in this section, for clarity. I would respectfully suggest wording along the following lines, for example: Aim 1. Assess whether children with CHD and EF impairments can adhere to the E-FIT program. Hypothesis 1. Study participants will complete at least 80% of the strategy coaching and computerized training sessions in the E-FIT program.
---

	Aim 2. Assess the acceptability of the E-FIT program for children with CHD and EF impairments and their parents. Hypothesis 2. Study participants and their parents will indicate high acceptability of the E-FIT program as measured by average ratings of 3 or higher on a 4-point rating scale assessing acceptability of each program component. Aim 3. Assess the feasibility of the E-FIT program for children with CHD and EF impairments and their parents. Hypothesis 3. Study participants and their parents will indicate high feasibility of the E-FIT program as measured by average ratings of 3 or higher on a 4-point rating scale assessing the user-friendliness and ease of integrating each program component into their daily lives and environments. Aim 4 (Exploratory). Measure the impact of the E-FIT program on improving EF skills in children with CHD and EF impairment. Hypothesis 4. We anticipate a moderate effect in the direction of improved EF from pre- to post-intervention in study participants. Otherwise, I have no further edits and look forward to seeing the results of this intervention study.
--	--

VERSION 2 – AUTHOR RESPONSE

Reviewer: 1 Dr. Marie Brossard-Racine, RI-MUHC Comments to the Author: The revised version read smoothly and the authors have addressed all my previous comments. This is very exciting work and I look forward to seeing the results of this study. Response: We thank Dr. Marie Brossard-Racine for providing valuable feedback on our manuscript and the positive feedback upon our revisions. We are delighted to hear of her excitement regarding our research. Her support is greatly appreciated, and we look forward to sharing our progress with her. Reviewer: 3 Silvia Bunge Comments to the Author: The authors have addressed all of my concerns. Response: We thank Prof. Silvia Bunge for her valuable feedback on our manuscript. We appreciate her feedback, and are pleased to hear that all of her concerns have been addressed to her satisfaction. Reviewer: 4 Kelly Wolfe Comments to the Author: I have one area of requested editing for this revised draft which is to further hone and refine the Specific Aims. Having very clearly defined Aims and Hypotheses will be crucial for evaluating the study's future results. The authors did re-work some of the Aims in response to my first revision, but the Aims should be refined further to focus only on empirical or measurable outcomes, and the fourth Aim is currently written as a hypothesis. I think it would be most useful to include both Aims and Hypotheses in this section, for clarity. I would respectfully suggest wording along the following lines, for example: Aim 1. Assess whether children with CHD and EF impairments can adhere to the E-FIT program. Hypothesis 1. Study participants will complete at least 80% of the strategy coaching and computerized training sessions in the E-FIT program. Aim 2. Assess the acceptability of the E-FIT program for children with CHD and EF impairments and their parents. 3 Hypothesis 2. Study participants and their parents will indicate high acceptability of the EFIT program as measured by average ratings of 3 or higher on a 4-point rating scale assessing acceptability of each program component. Aim 3. Assess the feasibility of the E-FIT program for children with CHD and EF impairments and their parents. Hypothesis 3. Study participants and their parents will indicate high feasibility of the E-FIT program as measured by average ratings of 3 or higher on a 4-point rating scale assessing the user-friendliness and ease of integrating each program component into their daily lives and environments. Aim 4 (Exploratory). Measure the impact of the E-FIT program on improving EF skills in children with CHD and EF impairment. Hypothesis 4. We anticipate a moderate effect in the direction of improved EF from pre- to post-intervention in study participants. Otherwise, I have no

further edits and look forward to seeing the results of this intervention study. Response: We thank Kelly Wolfe, PhD for her valuable input and suggestions for refining the specific aims of our study. We appreciate the careful consideration of this section, and agree that having clear and focused aims and hypotheses is essential for evaluating our study's results effectively. The suggested wording is very helpful, and was incorporated into the revised manuscript. We are delighted to hear of her interest regarding our research. Her support is greatly appreciated, and we look forward to sharing our progress with her.